# Enhancing Real-World Fall Detection Using Commodity Devices: A Systematic Study

**DOI:** 10.3390/s25175249

**Published:** 2025-08-23

**Authors:** Awatif Yasmin, Tarek Mahmud, Syed Tousiful Haque, Sana Alamgeer, Anne H. H. Ngu

**Affiliations:** 1Department of Computer Science, Texas State University, San Marcos, TX 78666, USA; nuc4@txstate.edu (A.Y.); sana.alamgeer@txstate.edu (S.A.); angu@txstate.edu (A.H.H.N.); 2Department of Electrical Engineering and Computer Science, Texas A&M University-Kingsville, Kingsville, TX 78363, USA

**Keywords:** fall detection in the real world, optimizing fall-detection model, fall detection with multiple sensors

## Abstract

The widespread adoption of smartphones and smartwatches has enabled non-intrusive fall detection through built-in sensors and on-device computation. While these devices are widely used by older adults, existing systems still struggle to accurately detect soft falls in real-world settings. There is a notable drop in performance when fall-detection models trained offline on labeled accelerometer data are deployed and tested in real-world conditions using streaming, real-time data. To address this, our experimental study investigates whether incorporating additional sensor modalities, specifically gyroscope data with accelerometer data from wrist and hip locations, can help bridge this performance gap. Through systematic experimentation, we demonstrated that combining accelerometer data from the hip and the wrist yields a model capable of achieving an F1-score of 88% using a Transformer-based neural network in offline evaluation, which is an improvement of 8% over a model trained solely on wrist accelerometer data. However, when it is deployed in an uncontrolled home environment with streaming real-time data, this model produced a high number of false positives. To address this, we retrained the model using feedback data that comprised both false positives and true positives and was collected from ten participants during real-time testing. This refinement yielded an F1-sore of 92% and significantly reduced false positives while maintaining comparable accuracy in detecting true falls in real-world settings. Furthermore, we demonstrated that the improved model generalizes well to older adults’ movement patterns, with minimal false-positive detections.

## 1. Introduction

Fall detection is a critical aspect of health monitoring, particularly for older adults at a higher risk of falls [1]. Early fall-detection systems relied on specialized hardware worn on the chest or waist to monitor sudden changes in movement and body orientation [2]. These early devices were often bulky and intrusive and restricted the free movement of users, leading to low acceptance among older adults. The increasing prevalence of smartphones and smartwatches equipped with multiple sensors and robust computational capabilities presents an opportunity to enhance fall-detection technologies. These devices, which are already widely adopted by older adults, offer a non-intrusive and portable solution for continuous health monitoring. However, the built-in IMU (Inertia Measurement Unit) sensors on smartwatches and smartphones are not of clinical-grade quality. As a result, the sensed data are of lower quality in that they can be noisy, with a less consistent sampling rate. Even when special care is taken to train machine learning models using clinical-quality sensed data acquired using watches like Empatica, their performance can degrade significantly when they are deployed on commodity devices due to the reduced accuracy of the real-time sensor input and the effect of the known model-drift problem on a trained model. The latter term refers to the phenomenon whereby an accurate model trained using data collected with a specific device gives sub-par performance when it is used in another device. Nonetheless, commodity-based devices are essential for enabling accessible, convenient, and portable solutions for long-term fall monitoring.

Recently, more and more systems for human-activity recognition have been implemented using various deep learning-based machine learning algorithms [3]. However, the data used for training the algorithms have still predominantly been accelerometer data due to privacy concerns associated with visual data and the general perception that edge devices lack the computing power to process data from multiple sources [4,5].

Despite advancements in machine learning, current fall-detection technologies deployed on commodity devices struggle with distinguishing Activities of Daily Living (ADLs) from soft falls, which are common among older adults. Most existing models rely heavily on accelerometer data from a single joint position alone; while this is practical for implementation, it limits the contextual information needed to distinguish between falls and ADLs. As documented in [6], there have been at least 57 projects that used wearable devices to detect falls in older adults. However, only 7.1% of the projects reported testing their models in real-world settings. Many papers in the literature provide very accurate fall-detection results in terms of both recall and precision via offline experiments, but those accuracies have not been verified in real-world settings.

Since there is a gap between the performance of the model trained offline using labeled data and the one deployed on a watch or phone using continuously streamed data, this paper aims to explore the use of additional sensor data, specifically gyroscope data from a device worn on the wrist, in conjunction with accelerometer and gyroscope data from a device worn on the opposite hip. We explore all possible combinations of these four data sources to see which combination is best for detecting both falls and ADLs using a dataset collected with Android-based commodity devices and tested in real-world settings with the same devices. The conventional wisdom is that having more sources of data leads to more accurate fall detection. But is there an advantage to using all sources, and what are the sources that can practically be used? To what extent do raw gyroscope-sensed data help to identify falls?

Through systematic experimentation, we demonstrated that combining accelerometer data from a hip and a wrist with a transformer neural network can significantly improve the model’s performance, achieving an F1-score of 88%. This represents an 8% increase compared to models using only the wrist’s accelerometer data. The experimental results also show that the addition of all types of data (gyroscope and accelerometer) from the two locations did not yield better performance compared to the accelerometer data from two opposing locations alone. This shows that not all additional contextual information from IMU sensors in commodity-based devices is useful. Note that the type of machine learning algorithm used and the quality and quantity of the sensed data all play a role in detection accuracy. However, finding the most effective deep learning-based algorithm to use is not the goal of this paper. We are interested in exploring just two joint positions that can be easily monitored with smartwatches and smartphones. For wide adoption of the system to be feasible, the fall-detection model must run on commodity-based devices. Therefore, extensive preprocessing to improve the quality of the sensed data is also not the emphasis of this paper. The limited amount of available fall data is a known problem in this research field that we have studied in a different paper [7].

Our work is different from others in the literature in that we aim to systematically explore the training of a fall-detection model that can be deployed on commodity-based devices and that has verified acceptable performance in the real world. We have built an Android-based SmartFall App for effective evaluation of real-world models [8]. The best-performing offline model is converted to a Tensorflow Lite model and installed on the SmartFall App. A two-stage user study protocol is designed to ensure consistent evaluation of the model by different participants in our lab (controlled environment) and in their homes (uncontrolled environment). The main contributions of this paper are as follows:We demonstrated the effectiveness of using the combined accelerometer sensor data from the wrist and the opposite hip for fall detection. We compared the accuracy of the model trained on different combinations of a sensor location and the sensor type and got the best F1-score using the accelerometer data from both the wrist and the opposite hip.We designed a two-stage user study protocol to evaluate the real-world performance of fall-detection model in a controlled environment (in the laboratory) and an uncontrolled environment (in the participant’s home).We demonstrated that incorporating the user feedback on wrongly classified participant data while using the system in controlled and uncontrolled environments can reduce the number of false positives by a large margin and is a viable approach to deploying a real-world fall-detection system for the targeted older adults.

## 2. Background

We analyzed the raw data in the SmartFallMM dataset [9], a multimodal dataset collected in our laboratory for the exploration of different modalities and data sources to improve fall detection.

To analyze the true-positive and false-positive cases, we visualized some of the accelerometer data collected from participants using the smartwatches and smartphones while they were performing prescribed ADLs and falls. Figure 1a and Figure 1b show a true-positive (right fall) and a false-positive (hand washing) sample, respectively. Specifically, in Figure 1a, the y-axis limit for fall events is approximately 15–20 ms^2^. Similarly, in Figure 1b, the y-axis limit for the hand-washing activity also reaches around 15–20 ms^2^, which gives it a significant resemblance to the fall data. This similarity in patterns between fall and ADL events sensed from a single wrist source presents a challenge for the model, as it complicates the task of accurately distinguishing between these activities.

Falls usually cause sudden and substantial movements at the hip, whereas common ADLs such as hand washing, waving, or drinking water mainly involve arm motions and produce very little movement around the hip. Because of this contrast, signals collected from the hip provide an additional layer of contextual information, making it easier to separate true falls from these everyday non-fall activities.

We visualized the hip’s accelerometer data to determine whether additional information could be extracted to better distinguish between ADL and fall activities. Figure 2a–d present the visualization of the same two activities that are depicted in Figure 1a and Figure 1b, respectively, but based on data generated from the hip rather than from the wrist. These figures illustrate how each sensor captures the same activity.

As shown in Figure 2a, which represents the hip’s accelerometer data, we observe a distinct spike in the data associated with the fall. However, the ADL activities, specifically hand washing, as shown in Figure 2d, are visualized, the hip’s accelerometer data display no notable spikes. We believe this variation in data between the wrist and hip sensors for ADL activities provides critical insights that could improve the accuracy of fall-prediction models.

We then plotted the gyroscope data for the same activities recorded from both the hip and the wrist. Figure 2b,c illustrate the gyroscope data from the wrist and hip, respectively, for a right fall. The data reflect specific movement patterns associated with the fall activity. Conversely, Figure 2e,f display the gyroscope data for the hand-washing activity. The wrist’s gyroscope data show some movement due to the significant hand activity involved in washing hands, whereas the hip’s gyroscope data remain relatively flat, indicating minimal hip movement during this activity.

## 3. Related Work

Sensor-based human-activity recognition (HAR) has evolved significantly over the past two decades, beginning with early systems like Transpose [10]. Among the key applications of HAR, fall detection has gained particular attention with the rise of wearable technologies. One of the early works is a study by Mauldin et al. [11] that evaluated fall-detection algorithms using raw accelerometer data from the SmartFall2018 dataset [12], where signals were sampled at 32 Hz and collected from seven volunteers wearing a Microsoft (MS) Band smartwatch on their left wrist. The study compared the performance of Naive Bayes (NB), Support Vector Machine (SVM), and a deep learning model based on Gated Recurrent Units (GRU). The GRU model achieved the highest F1-score (0.73) in a real-world test, with this value representing a notable drop from an F1-score of 0.87 in the offline test. This motivated a subsequent study [13] that explored the combination of ensemble learning approaches with user feedback to enhance real-world performance. However, ensemble learning approaches require substantial computational resources, making them unsuitable for deployment on resource-constrained edge devices. Additionally, the study did not explore the use of additional data sources, such as gyroscope signals, and additional sensor locations, such as the opposite hip.

Şengül et al. [14] proposed a fall-detection system based on accelerometer and gyroscope data. The system employed a Bidirectional Long Short-Term Memory (BiLSTM) neural network that was initially trained on data collected from 15 volunteers in a controlled environment using the Sony SmartWatch 3 (SWR50) at a sampling rate of 50 Hz. The trained model was deployed on a web server for activity classification. Raw sensor data were preprocessed, and 19 statistical features, including mean, minimum, maximum, variance, and entropy, were extracted separately from accelerometer and gyroscope signals, resulting in a 38-dimensional feature vector. When a fall was detected, a warning message was sent back to the smartphone. This entire process was evaluated in an offline setting, and the model achieved an F1-score of 0.99. The model was not tested in real-world conditions and relied on cloud-based inference, introducing latency and increasing power consumption through continuous data transmission.

Kulurkar et al. [15] developed a wearable sensor prototype that incorporated a Microelectromechanical Systems (MEMS) sensor module (LSM6DS0), and the sensed data were sent to the cloud using a 6LoWPAN service gateway. The MEMS sensor is a 3-axis accelerometer that collects movement data at a sampling rate of 50 Hz. The raw signals were preprocessed using a first-order Infinite Impulse Response (IIR) low-pass filter and used to train a model comprising a 1D convolutional neural network (CNN) layer followed by a long short-term memory (LSTM) layer (1DConvLSTM). Initial training was conducted offline using the MobiAct dataset [16]. The trained model was then deployed to the cloud. A total of six subjects were used for real-world testing, and the best performance was achieved using the data collected from sensor placements on the waist and wrist, with an F1-score of 0.96. The model was further retrained in the cloud each time a fall was detected, after which the updated model was redistributed to the local gateway. However, the performance before and after retraining was not reported. The system relies on specialized sensors not available on commodity-based smartphones and smartwatches.

Zhang et al. [17] proposed a fall-detection system based on accelerometer and gyroscope data, employing a dual-stream convolutional neural network with a self-attention mechanism (DSCS). The model achieved an offline F1-score of 0.95 on the MobiFall dataset [18] and maintained the same performance when deployed on a Huawei Watch 3 and tested in real time with 10 student participants. However, the model was trained on sensor data collected from waist-mounted devices, while the real-world evaluation was performed using a wrist-worn smartwatch. This discrepancy in sensor placement introduces a significant modality shift, which creates doubt in terms of the reliability and generalizability of the model in real-world scenarios with older adults.

Buzpınar [19] proposed a multisensor data-fusion approach to improve fall-detection accuracy. The study utilized two datasets [20]: the MTW-IMU dataset, collected with the Xsens MTw Awinda system, known for its effective drift-error reduction and high measurement precision [21], and the Activity Tracking Device (ATD) dataset, acquired using the BMX055 core sensor. The accelerometer and gyroscope data, sampled at 25 Hz, were used from both datasets. Ten machine learning models were evaluated on a hybrid dataset comprising 16 ADL classes and 20 fall classes. The Extra Trees classifier [22] achieved the highest offline F1-score, 0.99. However, the system was not tested in real time, leaving its performance after deployment unverified.

Similarly, Yhedgo [23] collected accelerometer and gyroscope data at a sampling rate of 200 Hz from human subjects using Noraxon myoMOTION IMU sensors [24]. A deep learning-based architecture was proposed for training a fall-detection system, integrating CNN, LSTM, and Transformer components, and achieved an offline F1-score of 0.96. However, the model was not tested in the real world. The reliance on high-end IMUs with a 200 Hz sampling rate limits the generalizability of the results to real-world conditions, where commodity sensors typically operate at much lower sampling rates.

TinyML [25] and federated learning [26] represent promising directions in the broader area of on-device, privacy-preserving human-activity recognition and health monitoring. However, our study specifically targets commodity smartwatches and smartphones as deployment platforms. These devices already support TensorFlow Lite (TFLite) inference with significantly greater computational resources than the microcontroller-class hardware typically considered in TinyML research.

In summary, Table 1 highlights that many existing fall-detection methods depend on handcrafted features, prefiltered signals, or high-precision sensors. Although these systems often report high F1-scores, their evaluations are primarily conducted offline on limited simulated datasets. Some methods have employed specialized sensors. However, their performance is not guaranteed when these sensors are embedded in consumer devices due to issues like noise, sampling instability, and limited firmware control. While the use of the combination of gyroscope and accelerometer data is popular in many approaches, the contribution of gyroscope data is not quantified. While our method also uses simulated data for training, it distinguishes itself by leveraging raw signals from commodity devices and validating performance in the real world, including in the targeted older adult population.

## 4. Methodology

### 4.1. Datasets

In this study, we utilized a subset of the SmartFallMM [9], which consists of both fall and ADL data. The subset of data was obtained from 16 student participants (11 males and 5 females) aged 23 to 33 years. The dataset contains data from two inertial sensors, specifically an accelerometer and a gyroscope, and were collected from a smartwatch positioned on the left wrist and a smartphone securely positioned on the right hip within a harness. The data consist of five distinct types of falls (front, back, left, right, and rotational). Each fall type was executed five times onto an air mattress, and each ADL was also repeated five times. The types of ADL activities include walking, waving hands, drinking water, wearing a jacket, sitting/standing, washing hands, picking objects up from the floor, and sweeping the floor. In total, we have 400 fall samples and 640 ADL samples. These data were collected with the approved Institutional Review Board (IRB) numbers 7846 and 9461.

### 4.2. Data Preprocessing

The only data preprocessing involved segmenting the time-series data into chunks that could be fed to the machine learning model. We employed the sliding-window technique to segment data into a series of overlapping windows. Through experimentation with various window sizes, we selected a window size of 128 data points, as it yielded better results in distinguishing between falls and ADLs in our earlier work [27]. Given the sampling rate of 32 ms of the smartwatch used in our studies, each window encompassed around 4 s of activity.

Model predictions (i.e., predictions produced by the deep learning algorithm) begin once the number of data points acquired from the sensor is equal to the configured window size. Every model prediction thereafter will require additional data points only as specified by the stepSize. The model will slide *n* data points, as specified by the stepSize, at a time, reusing all of the previous data points except for the least recent. However, before producing a final prediction, the model will generate a heuristic value based on the probabilities produced by several consecutive predictions. In essence, the model computes the average value of ten consecutive probabilities and compares this with a predefined threshold value, which is set at the default value of 0.5. If the average probability exceeds this threshold, then it is considered a fall prediction. This helps to prevent isolated positive model predictions from triggering a false positive.

### 4.3. Computation Model

We have conducted a comparative analysis of different model architectures for fall detection using various datasets, including the SmartFallMM from [27]. The transformer model performed best and thus was chosen for our systematic experimentation. The architecture of the transformer we used is shown in Figure 3. It features four encoder layers, each equipped with four attention heads; this design choice was aimed at balancing the model size and performance metrics for deployment on resource-constrained wearable devices. Given that our task does not involve generating new sequences, the decoder portion in the original Transformer architecture is omitted. The attention mechanism within the Transformer model is pivotal.

Multihead attention (MHA) operates by mapping queries and key–value pairs to an output. Within a single attention head, scaled dot products of queries (Q) and keys (K) are computed and scaled by the dimensionality of the keys $d_{k}$. This process is followed by softmax normalization to derive weights, which are subsequently multiplied by values (V), as depicted in Equation (Equation 1). The outputs from multiple attention heads are concatenated and projected using projection matrix $W^{O}$ to obtain the final output, where ${\mathrm{head}}_{i}$ is the output of each head of MHA, as illustrated in Equation (Equation 2).

Layer normalization, a technique intrinsic to the Transformer architecture, normalizes the activation of each layer independently, thereby stabilizing training and addressing issues such as vanishing gradients and internal covariate shift. This normalization technique promotes gradient flow, accelerates convergence, and reduces sensitivity to hyperparameters, ultimately contributing to the model’s effective training and enhanced performance.

The final layer of the Transformer encoder comprises a feed-forward module (FFN), which processes the output of MHA through weight matrices $W_{1}$ of dimension eight and a ReLU activation to later process it with another weight matrix $W_{2}$ of dimension 16, as described in Equation (Equation 3). This output goes through a prediction layer with one neuron to predict falls and ADLs. The sigmoid function serves as the activation function of the prediction/output layer, yielding probabilities between zero and one, and the binary cross-entropy loss function was used for training the model. For all the experiments, we used a Transformer architecture with four encoder layers and four MHA heads, the embedding size for the attention layer was fixed to 128, and the dropout rate was set to 0.25 to counter overfitting.(1)$$\mathrm{Attention}\left(Q,K,V\right)=\mathrm{softmax}\;\left(\frac{QK^{⊤}}{\sqrt{d_{k}}}\right)V$$(2)$$\begin{array}{cc} \mathrm{MHA}(Q,K,V) & =\mathrm{Concat}\left({\mathrm{head}}_{1},\ldots ,{\mathrm{head}}_{h}\right)W^{O},\\ \mathrm{where}\;{\mathrm{head}}_{i} & =\mathrm{Attention}(Qi,K_{i},V_{i}),\;i=1,\ldots ,h \end{array}$$(3)$$\mathrm{FFN}\left(\mathrm{MHA}\left(Q,K,V\right)\right)=\max\;\left(0,\;\mathrm{MHA}\left(Q,K,V\right)W_{1}+b_{1}\right)W_{2}+b_{2}$$

### 4.4. Experiments

We first evaluated the offline machine learning model, where we trained the transformer model with all possible combinations of both accelerometer and gyroscope data collected from the wrist and the hip. We focus only on the left-wrist and right-hip locations because our target users are older adults who are less likely to accept a system that requires attaching too many sensors to their bodies.

Given the limited size of the dataset, we implemented a Leave-One-Out cross-validation strategy. We trained the models using the TensorFlow toolkit on a Dell Precision 7820 Tower with 256 GB RAM and one GeForce GTX 1080 GPU. We ran each experiment three times to check the consistency of the model. To evaluate and compare the performance of different models, the standard performance metrics of F1-score, precision, recall, and accuracy are used.

The best-performing offline model (the one with the highest F1-score) out of all the combinations of data sources was chosen for the first real-world evaluation, which was conducted in a controlled environment. In this experiment, the participants were instructed to perform five types of falls onto a queen-size air mattress and to engage in eight prescribed ADLs in a closed room in our laboratory. Each ADL was repeated five times to ensure consistency and to assess the model’s initial precision and recall.

The second real-world evaluation was conducted in the participants’ homes, an uncontrolled environment, where they used the fall-detection application running on a smartphone-smartwatch pair. Participants went about their normal daily activities for three hours per day for three days. The participant’s only task was to provide feedback to the application in case of any false-positive or true-positive triggers. This approach allowed us to assess the model’s performance and adaptability in real-world conditions. This user feedback and the corresponding data were saved in Couchbase, a non-SQL database. On the fourth day, we downloaded the user-feedback data (from all users participating in the study), combined them with the original dataset, and retrained a new model. This retrained model was deployed to the devices, which were then given to the participants for use over the next three days.

For both controlled and uncontrolled real-world experiments, ten students were recruited (all students had to sign consent forms as part of the IRB-approved protocols 7846 and 9461). The consent form detailed the purpose of the study, the tasks required for participation, and any potential risks or discomforts. Each participant wore the smartwatch on their left wrist and secured the smartphone in a harness on their right hip during the experiments.

#### The SmartFall App

To deploy a specific trained model on the smartwatch, we utilized the Tensor Flow Lite (TFLite) model format supported by the Andriod OS system. The trained offline model was saved in the TFlite format and deployed on the SmartFall App.

Figure 4 illustrates the user interface of the SmartFall App on both the smartphone and the smartwatch used for the real-world evaluation. Once the devices are paired, pressing “ACTIVATE” on the phone’s UI initiates continuous sensing of data from the embedded sensors on the smartwatch and smartphone. After that, all interactions happen through the smartwatch, which initially displays the “FELL” screen. The system makes a prediction after 128 sensor data points have accumulated from both smartphone (hip accelerometer) and smartwatch (wrist accelerometer).

When a fall is detected, the system shows the “Did you Fall?” screen with “YES” and “NO” option buttons to fit the small screen space, as shown in Figure 4. If the user selects “NO”, indicating a false alarm, the data are labeled and saved as “FP” (false positive). If the user selects “YES”, the system then asks, “NEED HELP?”. If the user answers “YES”, it confirms a true fall and the need for help. Then, it displays the “HELP IS ON THE WAY” screen. The data are labeled and saved as “TP” (True Positive). If the user answers “NO” to needing help after detection of a true fall, it also labels and saves the data as “TP” and goes back to the first “FELL” screen. This interaction allows the system to count false positives, true positives, false negatives, and true negatives to compute the real-world performance metrics. These labeled data are then sent to Couchbase for storage. We update the locally stored data once the user provides the feedback. The locally stored data with user feedback is uploaded to a cloud-based server on a preconfigurable interval (15 min in this experiment) for archival. For the controlled experiments, participants were instructed to perform five types of falls onto an air mattress and engage in eight types of Activities of Daily Living (ADL) in the laboratory. Each activity was repeated five times to ensure consistency and to assess the model’s initial precision and recall. All our controlled and uncontrolled real-world experiments with subjects were approved under Texas State University IRB numbers 7846 and 9461. In this study, we used a Google Pixel 6 phone with 128 GB of memory and a TicWatch Pro 3 smartwatch to perform all real-world evaluation experiments.

## 5. Results

### 5.1. Offline Evaluation

For our first experiment, i.e., the offline experiment, we trained four single-location and single-sensor (SLSS) experiments as baseline models to compare with the multi-location and multi-sensor-based models (MLMS) using Transformer. The SLSS combinations include Wrist Accelerometer (wA), Hip Accelerometer (hA), Wrist Gyroscope (wG), and Hip Gyroscope (hG). On the other hand, the MLMS combinations include wA+wG, hA+hG, wA+hA, wG+hG, wA+hG, hA+wG, wA+hA+wG, wA+hA+hG, wA+wG+hG, hA+wG+hG, and wA+wG+hA+hG. Please note that, to form each combination, we horizontally stacked the data from the selected modalities. For example, combining Wrist Accelerometer (wA) and Hip Accelerometer (hA), each with shape 1×3, resulted in a single 1×6 dimensional input vector.

Table 2 shows the Precision, Recall, and F1-score of the four baseline models. The recall directly relates to the system’s ability to detect true falls, which is critical for ensuring user safety, while precision indicates how often the detected falls are indeed real falls, which is important for minimizing false alarms and preventing alarm fatigue. The F1-score provides a balanced view of both aspects. By explaining these connections, we ensure that the reported metrics are better contextualized for real-world deployment and usability. The model trained with the wA data alone achieved a precision of 0.79, a recall of 0.84, and an F1-score of 0.81. In comparison, the model trained with the hA data showed a lower precision of 0.67 but a higher recall of 0.94, resulting in an F1-score of 0.78. The highest recall (0.94) indicates that the hA is better at identifying true positives (falls) than wA.

As most falls involve certain hip movements, the hA can identify most of the falls correctly. However, certain ADL activities (such as sitting and standing, picking up objects, sweeping, etc.) also involve significant hip movements, which results in their misclassification. As shown in Table 2, the baseline model trained solely on gyroscope data indicates that the model using wG data achieved a precision of 0.76, outperforming any accelerometer-based models. However, its recall was lower, at 0.60. This suggests that wG is effective in correctly identifying activities of daily living (ADLs) but tends to miss a significant number of falls. In contrast, the model trained on hG data had the lowest precision, at 0.59, but a higher recall (0.81) compared to wG. This means that while the hG performs better at detecting falls, it struggles with accurately identifying ADL activities. These observations highlight that each sensor has unique strengths, with the wA achieving the best balance between precision and recall, making it the most suitable standalone sensor for fall detection. From this point onward, we refer to this trained model as SLSS-I.

Table 3 shows the results from training models with all possible combinations of MLMS. First, we combined accelerometer and gyroscope data from a single location. The model trained on wA+wG data achieved a high recall of 0.98 but a low precision of 0.63. For the hip location, the model trained on hA+hG data had a precision of 0.76, which was an improvement over the wrist model, but its recall was 0.78; this was low compared to the hA alone, which had a recall of 0.94. These results suggest that combining accelerometer and gyroscope data from either the wrist or the hip does not significantly improve the model’s ability to distinguish between falls and activities of daily living (ADLs).

Next, we combined data from different sensor locations. Specifically, wA+hA data yielded the highest F1-score so far for the model, at 0.88. Training the models with wG+hG data resulted in a lower F1-score of 0.75, while wA+hG data provided a balanced result for the model, with a precision of 0.82 and a recall of 0.86, surpassing the results from single-location data.

Last, hA+wG data resulted in a high precision (0.92) for the model but a lower recall (0.68). While this combination improved ADL identification, the model struggled to effectively differentiate between falls and ADLs, making it less suitable for fall detection.

After that, we added gyroscope data to the best multi-location sensor model, which is wA+hA. First, we added wrist gyroscope data to form the wA+hA+wG combination. This combination resulted in a recall of 0.96 and a precision of 0.72, with a precision lower than that of the model that excluded gyroscope data. Next, the combination wA+hA+hG led to an even higher recall of 0.98, but this came at the cost of a further drop in precision to 0.71.

We then tested combinations of one accelerometer with gyroscopes from both locations. The model trained with the hip accelerometer and both gyroscopes (hA+wG+hG) achieved perfect recall (1.00) but had the lowest precision, at 0.59. Finally, when all four sensors were combined (wA+wG+hA+hG), the result was a high recall of 0.97 but a low precision of 0.69.

This analysis indicates that combining accelerometer data from two locations (hip and wrist) enhances the model’s ability to detect falls and ADLs. The use of raw gyroscope data does not improve model accuracy [28]. Adding more gyroscope data from multiple locations adds noise, making it more challenging for the model to detect falls accurately. Accelerometer data can yield better performance, as they capture dynamic changes in linear motion. They measure ground reaction forces, which are stronger on firm surfaces. This gives more information about joint loading and movement. As a result, the model using accelerometer data from both locations has the best F1-score (0.88). From this point onward, we refer to this initial, offline-trained model as MLMS-I.

### 5.2. Real World Evaluation

In real-world evaluation, we compared the best-performing SLSS-I with the best-performing MLMS-I (see Table 2 and Table 3), and the results are summarized in Table 4.

In the first real-world evaluation in the controlled environment, the average precision and recall values show that the model MLMS-I performed better compared to the model SLSS-I when it was used by ten participants in the laboratory. Table 4 compares the performance of these two models. The average F1 score improved by 7%; the recall improved by 3%; and the precision improved by 11%. Adding data from both the wrist and the hip’s accelerometers improved precision, recall, F1-score, and accuracy for most of the users. The percentage improvement or decline for each participant is shown in the table inside the parentheses. However, as seen in Table 4, precision, though improved, remained below 0.80 for most users.

We then instructed the same ten participants to use the system at home for three days in an uncontrolled environment. In this second real-world evaluation in the uncontrolled environment, the MLMS-I showed a high rate of false positives. Activities like going down stairs or lifting heavy objects, which were not in the training dataset, were often misclassified as falls.

The ADLs in the SmartFallMM dataset comprised eight distinct activities, some involving only wrist movements and others involving full-body movements. Activities such as washing hands, wearing a jacket, waving hands, and drinking water involve only minimal hip movement. This explains why incorporating hip accelerometer data alongside wrist data during training improved the model’s ability to distinguish between them. However, the model trained with wA+hA data still struggles with false positives when the user is walking, especially when turning around with high intensity.

## 6. User-Feedback Integration

### 6.1. Methodology

The real-world evaluation in an uncontrolled environment showed that the MLMS-I model still has too many false positives for adoption by users. The objective of this experiment is to check whether integration of the feedback given by users while using the system can mitigate this false-positive problem. We collected four types of data from participants: true positives (TP), false positives (FP), true negatives (TN), and false negatives (FN). Then, we added these labeled data to the original SmartFallMM dataset for retraining. After all data were combined, the dataset included a total of 650 falls (TP and FN) and 23,179 ADL samples (TN and FP). If the model failed to recognize a fall, the user still had the option to step in and press the “FELL” button on the app’s home screen (Figure 4). This ensured that the fall was not ignored and was properly logged in the system as a false negative (FN). In our real-world evaluation, we treated these user-reported FN events as equal in value to the system’s true-positive (TP) detections. To strengthen the model, the FN and TP samples were merged and used during retraining. Thus, the model was exposed not only to the falls it had correctly identified but also to the ones it had initially missed. Over time, this feedback loop helps the system adapt better to different types of falls and user movements, making it more reliable in real-world conditions.

To evaluate the impact of these collected data on improving model accuracy, we retrained four distinct models. Three models were trained by adding all types of data (TP, FP, and TN) along with the initial SmartFallMM dataset, but with varying amounts of TNs: all (650 falls and 23,719 ADL samples), half (650 falls and 11,590 ADL samples), and a quarter of the TNs ( 650 falls and 5795 ADL samples), as the majority of data collected is TN. The fourth model was trained exclusively by adding only the user-feedback data (TP and FP) to the initial dataset, for a total of 650 falls and 1040 ADL samples. This approach allowed us to assess the contribution of user-feedback data, as well as that of the true negatives, to the overall performance of the models. All models were trained using the transformer architecture, following the methodology described in Section 4.

The best-performing offline model was selected for another round of real-world evaluation. The same group of 10 participants who had been involved in evaluating the initial model were asked to evaluate the best retrained model. They completed both controlled and uncontrolled real-world evaluations, following the same procedures as before.

Finally, we recruited eight older participants (ages greater than 60) to perform a test in an uncontrolled environment with both the initial and retrained models to assess the generalizability of the model trained on the collected data.

### 6.2. Evaluation with Retrained Model

Table 5 shows that the model retrained after only the feedback data (TP and FP) were added achieved the best recall, at 0.93, and an F1-score of 0.88. We refer to this retrained model as MLMS-II. However, the retrained model trained on all the data collected from the ten participants achieved a very high precision of 0.99 but a much lower recall of 0.54. This drop in recall happened because the dataset was highly unbalanced, with far more ADL samples than fall samples. The model trained with half of the true-negative data, along with the feedback and initial data, achieved the same precision of 0.99 but slightly better recall, at 0.62. The model trained with one-fourth of the TN data, including feedback and initial data, had slightly lower precision at 0.98 but slightly higher recall, at 0.63. This indicates that the imbalance in the data biases the model toward ADL activities, leading to high precision but low recall. Therefore, we selected the MLMS-II, model for further real-time evaluation to compare its performance with that of the original model.

Table 6 shows a performance comparison of MLMS-I and MLMS-II models after retraining with the feedback data across metrics: Precision, Recall, and F1-score. Each participant’s percentage improvement (+) or decline (−) is annotated inside the parenthesis. The table highlights consistent performance improvements with retraining, particularly in Precision and F1-score, demonstrating the effectiveness of incorporating user-feedback data. The MLMS-II model performed better, specifically in identifying the activities that were in the false-positive list before. The average recall was improved by 10%, and the precision was improved by 13%. This indicates better performance in detecting ADLs while maintaining accuracy in distinguishing between falls and non-fall activities.

When the MLMS-II model was evaluated in the uncontrolled environment, all ten users reported very few false positives. On average, they received a maximum of two false positives in an hour. This indicates that precision has improved, while high recall is maintained. This demonstrates that user-feedback data significantly improves the model’s precision and helps the model better distinguish between falls and ADL activities.

### 6.3. Evaluation with Older Adults

To evaluate the generalizability of the models to older people, we recruited eight older participants (five female, three male) aged between 60 and 92 years under the same IRB. Each participant used the system at home (uncontrolled environment) for 3 h a day over 3 days, and they were advised to complete three online surveys and a post-survey interview. Four participants used the MLMS-I from Table 4, which was trained on fall and ADL data from younger individuals, resulting in average daily FP counts of 15, 19, 20, and 21, respectively. The remaining four participants used the MLMS-II model from Table 6 with FP counts of two, four, two, and three. Notably, one participant reported experiencing three near falls, all of which were correctly detected by the app. These results suggest that while models trained on data from younger individuals can provide a reasonable starting point for fall detection in older adults, their performance improves significantly with retraining on feedback data. Results also show that this strategy can be a promising approach to enhance model generalizability and reliability for older users in real-world settings, though further validation on a larger sample is necessary.

In considering the practical applicability of our system, requiring users to wear a smartwatch and carry a smartphone may impact overall usability. To assess this, we conducted surveys and post-study interviews with eight older adult participants, focusing on their willingness to use the system under these conditions. Seven participants expressed that carrying a smartphone near the hip was inconvenient. However, all participants indicated a strong preference for a more discreet solution, such as a coin-sized sensor that could be attached to the body, highlighting their interest in the benefits of the system, including continuous monitoring and the ability to alert caregivers in the event of a fall, particularly at home or during outdoor activities like gardening.

## 7. Discussion

In our study, false positives were primarily associated with individual differences in gait patterns and motion signatures. Daily activities such as abrupt arm movements, rapid posture changes, or vigorous ADLs often generated sensor signals resembling those of a fall. Since each person exhibits unique movement dynamics, the same action could be considered normal for one individual but misclassified as a fall for another, highlighting the difficulty of generalizing across diverse users. To address this challenge, we incorporated user feedback into the framework, allowing the system to learn from misclassifications and adapt progressively to individual motion characteristics. This feedback-driven adaptation strengthened robustness and reduced the recurrence of false positives in real-world settings.

Our findings demonstrate that combining accelerometer data from the wrist and hip provides the most effective sensor configuration for fall detection on commodity devices, achieving an F1-score of 0.88. By contrast, adding gyroscope data did not yield improvements and sometimes reduced precision, showing that the use of more sensing modalities does not necessarily enhance performance. This underscores the need to prioritize complementary sensor placements rather than maximizing sensor count.

Real-world evaluations confirmed that while the dual-accelerometer model outperformed single-sensor setups in controlled environments, it produced more false positives in uncontrolled home settings, especially for activities not represented in training. Incorporating user feedback into retraining addressed this limitation: the refined model (MLMS-II) achieved an F1-score of 0.92, significantly reducing the number of false positives while maintaining high recall. Evaluations with older adults further validated this approach, as the retrained model reduced the number of daily false alarms to two–four while successfully detecting near-falls. These results highlight how accelerometer fusion and feedback integration improve not only accuracy, but also generalizability to older populations.

Although several fall-detection datasets exist (e.g., FARSEEING [29], Aziz et al. [30], Bagala et al. [31], Harari et al. [32], CareFall [33]), none include the exact combination of modalities and placements-simultaneous wrist and hip-mounted accelerometers and gyroscopes used in this study. This limits their direct applicability for evaluating our framework. By systematically analyzing commodity wearable devices in both controlled and real-world conditions, our work provides insights into practical sensor configurations that balance accuracy, usability, and deployability. These findings emphasize the benefits of carefully chosen sensor placements and highlight that thoughtful fusion strategies can outperform maximal sensor integration.

## 8. Conclusions

In this paper, we presented a systematic experimentation of different sources and a combination of sensor data to improve the fall-detection model on commodity devices. The experimental results demonstrate that using a combination of accelerometer data from the smartwatch and smartphone with a Transformer network yielded the highest F1 score (0.88) in our offline testing. While 0.88 might not look like a high accuracy number, the devices we used to sense and test the system are low-cost commodity devices practical for wide adoption. Moreover, we introduced protocols to evaluate the performance of our offline model in real-world scenarios in both controlled and uncontrolled environments. We demonstrated that adding the additional user feedback on true positives and false positives data and retraining a new model results in a retrained model that generates many fewer false positives without affecting the recall. This work paves the way for personalized fall-detection models on commodity devices, enhancing real-world applicability. We plan to investigate personalized model tuning to adapt to individual movement characteristics and improve detection accuracy. To enable such personalization while preserving user privacy, we aim to explore federated learning approaches. Finally, larger-scale studies with older adults in real-world conditions will be pursued to validate the robustness and clinical relevance of the proposed framework.

## Figures and Tables

**Figure 1 sensors-25-05249-f001:**
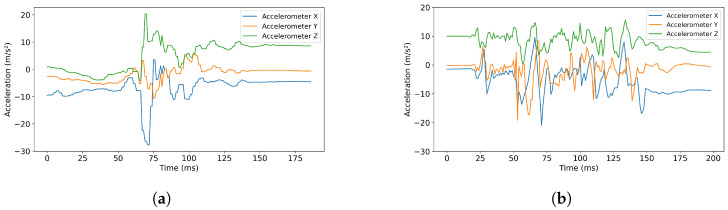
Wrist accelerometer for a right fall & washing hands. (**a**) Wrist accelerometer for a right fall (true positive). (**b**) Wrist accelerometer for washing hands (false positive).

**Figure 2 sensors-25-05249-f002:**
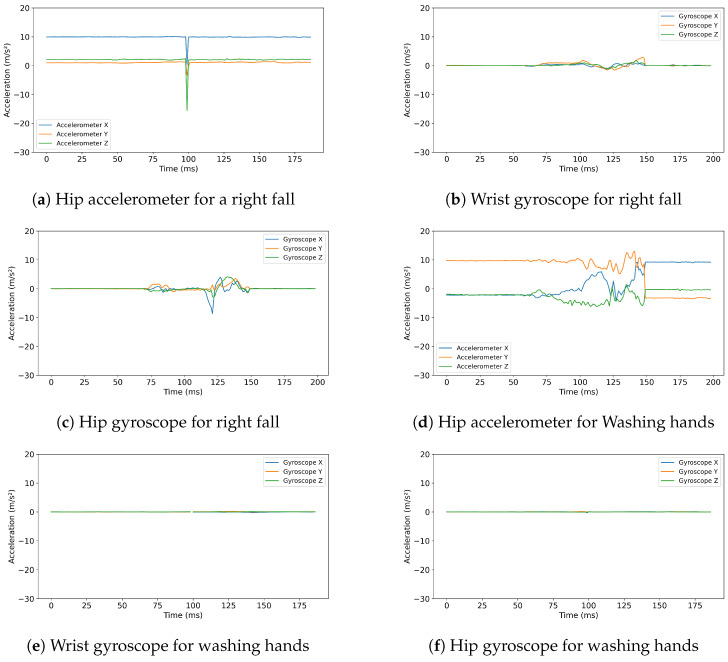
Hip accelerometer, wrist & hip gyroscope for a right fall & washing hands.

**Figure 3 sensors-25-05249-f003:**
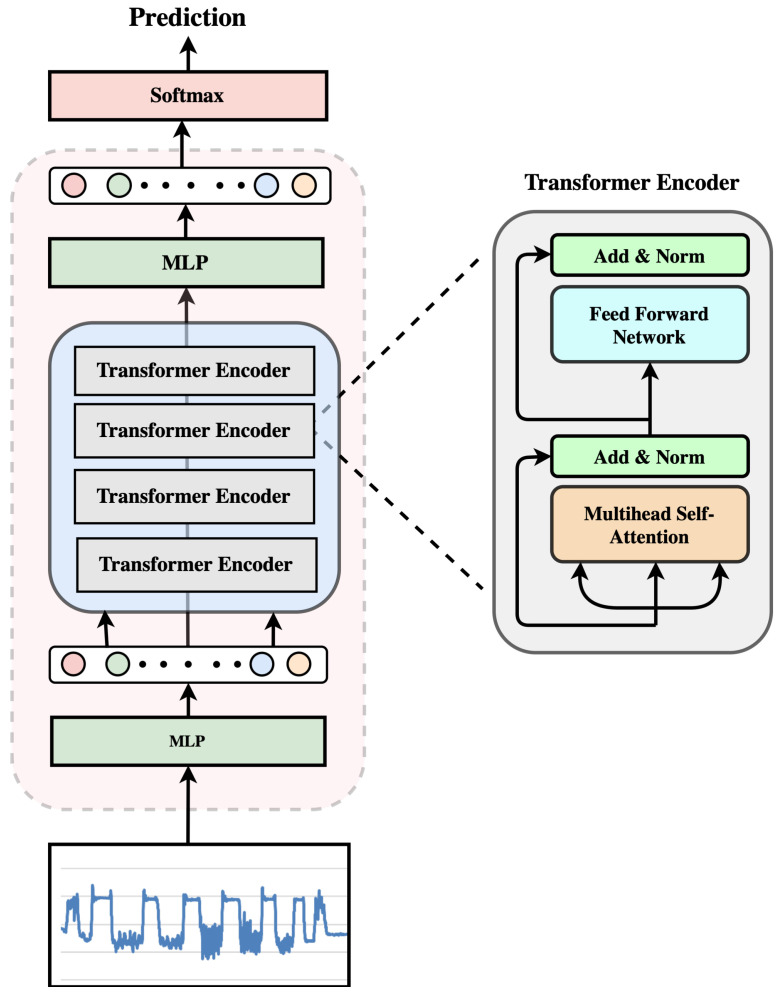
Transformer architecture.

**Figure 4 sensors-25-05249-f004:**
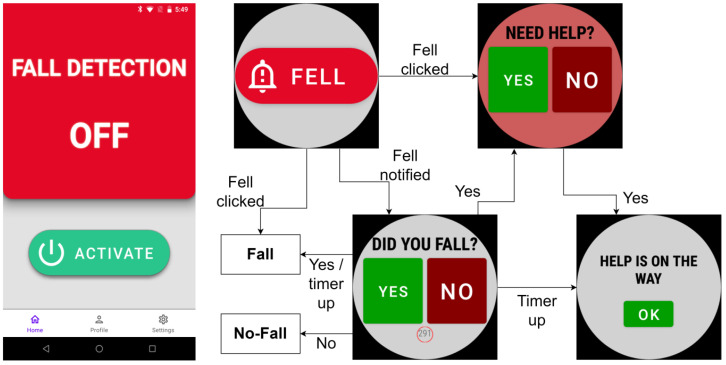
Fall-detection application.

**Table 1 sensors-25-05249-t001:** Comparison of fall-detection approaches using wearable sensors.

Paper	Placement/Sensors/Sampling Rate	Feature Extraction	Model	Device Name/Type	Dataset/Activities	Test Type/F1-Score	Limitations
Mauldin et al. [11]	Left Wrist/Accelerometer/32 Hz	No	GRU	Microsoft (MS) Band smartwatch/Commodity	SmartFall2018 [12]/ADLs: jogging, sitting, waving, walking. Falls: front, back, left, right	Real-world/0.73	(1) Relied on data from a single sensor only. (2) F1-score dropped in real-world testing from 0.87 to 0.73. (3) MS Band is discontinued.
Şengül et al. [14]	Left Wrist/Accelerometer +Gyroscope/50 Hz	Yes	BiLSTM	Sony SmartWatch 3 SWR50/Commodity	Dataset not available/ADLs: sitting, squatting, walking, running. Falls: while walking, from a chair	Offline/0.99	(1) Not tested in real-world, and relied on cloud-based prediction, causing latency (2) Limited fall diversity (only 2 fall types), reducing generalizability (3) The device runs outdated Android Wear 1.5 with no updates
Kulurkar et al. [15]	Waist+Wrist/Accelerometer/50 Hz	No	1DConvLSTM	LSM6DS0/ Specialized Sensor	MobiAct [16]/ADLs: 9 different classes. Falls: 4 different classes	Real-world/0.96	(1) Relied on cloud-based inference, increasing latency (2) Used IIR low-pass filter, which may suppress sharp fall signals (3) Data collected and tested in different positions (4) The initial LSTM model trained on waist data raises concerns about its generalizability to wearable sensors on other body locations.
Zhang et al. [17]	Left Wrist/Accelerometer +Gyroscope/87–200 Hz	No	Two-stream CNN with Self-Attention	Huawei Watch 3/Commodity	MobiFall [18]/ADLs: 9 different classes. Falls: 8 different classes	Real-world/0.95	(1) Trained using waist-mounted data, tested on wrist-worn device, creating modality mismatch (2) IMU devices used high sampling rate, not available in commodity watches
Buzpinar [19]	Waist/Accelerometer +Gyroscope/25 Hz	No	Extra Trees Classifier	Xsens MTWAwinda, andATD-BMX055/Specialized Sensor	MTW-IMU and ATD [20]/ADLs: 16 different classes. Falls: 20 different classes	Offline/0.99	(1) Not tested in real-world scenarios (2) Used high-precision data [21] for training, which are not available from commodity devices (3) Results may not generalize to smartwatches or phones used in real-world scenarios deployments
Yhedgo [23]	Shank/Accelerometer +Gyroscope/200 Hz	Yes	Model with CNN, LSTM, and Transformer components	NoraxonmyoMOTION/Specialized Sensor	Dataset not available/ADLs: unspecified. Falls: near-fall, forward, backward, obstacle	Offline/0.96	(1) Not tested in real-world scenarios (2) Dataset details unclear, fall and ADL class diversity not reported (3) Used specialized IMUs with a high sampling rate, not representative of consumer devices

**Table 2 sensors-25-05249-t002:** Model performance across single-location and single-sensor (SLSS) training configurations. The bold value indicates the best-performing model (SLSS-I).

Training Data	Precision	Recall	F1 Score
Wrist Accelerometer (wA)	0.79	0.84	**0.81**
Hip Accelerometer (hA)	0.67	0.94	0.78
Wrist Gyroscope (wG)	0.76	0.60	0.67
Hip Gyroscope (hG)	0.59	0.81	0.68

**Table 3 sensors-25-05249-t003:** Model performance across multi-location and multi-sensor (MLMS) training configurations. The bold value indicates the best-performing model (MLMS-I).

Training Data	Precision	Recall	F1 Score
wA+wG	0.63	0.98	0.76
hA+hG	0.76	0.78	0.77
wA+hA	0.87	0.92	**0.88**
wG+hG	0.76	0.73	0.75
wA+hG	0.82	0.86	0.84
hA+wG	0.92	0.68	0.78
wA+hA+wG	0.72	0.96	0.82
wA+hA+hG	0.71	0.98	0.82
wA+wG+hG	0.71	0.75	0.73
hA+wG+hG	0.59	1	0.74
wA+wG+hA+hG	0.69	0.97	0.81

**Table 4 sensors-25-05249-t004:** Real-world evaluation results for 10 participants using SLSS-I versus MLMS-I models.

Participant	Model	Precision	Recall	F1 Score
Participant 1	wA	0.58	0.84	0.69
	wA+hA	0.71 (+0.13)	0.88 (+0.04)	0.79 (+0.1)
Participant 2	wA	0.62	0.84	0.71
	wA+hA	0.78 (+0.16)	0.84 (+0)	0.81 (+0.1)
Participant 3	wA	0.7	0.84	0.76
	wA+hA	0.88 (+0.18)	0.92 (+0.08)	0.9 (+0.14)
Participant 4	wA	0.64	0.72	0.68
	wA+hA	0.72 (+0.08)	0.84 (+0.12)	0.78 (+0.1)
Participant 5	wA	0.69	0.78	0.73
	wA+hA	0.76 (+0.07)	0.8 (+0.02)	0.78 (+0.05)
Participant 6	wA	0.7	0.84	0.76
	wA+hA	0.75 (+0.05)	0.88 (+0.04)	0.81 (+0.05)
Participant 7	wA	0.73	0.78	0.75
	wA+hA	0.78 (+0.05)	0.71 (−0.07)	0.75 (+0)
Participant 8	wA	0.71	0.79	0.75
	wA+hA	0.77 (+0.06)	0.8 (+0.01)	0.78 (+0.03)
Participant 9	wA	0.69	0.72	0.71
	wA+hA	0.79 (+0.1)	0.76 (+0.04)	0.78 (+0.07)
Participant 10	wA	0.66	0.84	0.74
	wA+hA	0.81 (+0.15)	0.84 (+0)	0.82 (+0.08)
Average	wA	0.67	0.8	0.73
	wA+hA	0.78 (+0.11)	0.83 (+0.03)	0.8 (+0.07)

**Table 5 sensors-25-05249-t005:** Offline results from retraining the MLMS-I model using variable-length segments of user feedback. The bold value indicates the best-performing model (MLMS-II).

Training Data	Precision	Recall	F1-Score
Initial dataset + TP + FP	0.84	0.93	**0.88**
Initial dataset + TP + FP + TN	0.99	0.54	0.70
Initial dataset + TP + FP + 1/2 TN	0.99	0.62	0.76
Initial dataset + TP + FP + 1/4 TN	0.98	0.63	0.77

**Table 6 sensors-25-05249-t006:** Results from evaluating the MLMS-II model with the same group of 10 participants.

Participant	Precision	Recall	F1 Score
Participant 1	0.96 (+0.25)	0.92 (+0.04)	0.94 (+0.15)
Participant 2	0.86 (+0.08)	0.96 (+0.12)	0.91 (+0.1)
Participant 3	0.92 (+0.04)	0.96 (+0.04)	0.94 (+0.04)
Participant 4	0.93 (+0.21)	1 (+0.16)	0.96 (+0.18)
Participant 5	0.95 (+0.19)	0.92 (+0.12)	0.94 (0.16)
Participant 6	0.89 (+0.14)	0.96 (+0.08)	0.92 (+0.11)
Participant 7	0.91 (+0.13)	0.84 (+0.13)	0.87 (+0.12)
Participant 8	0.91 (+0.14)	0.84 (+0.04)	0.87 (+0.09)
Participant 9	0.91 (+0.12)	0.92 (+0.16)	0.92 (+0.14)
Participant 10	0.89 (+0.08)	0.96 (+0.12)	0.92 (+0.1)
Average	0.91 (+0.13)	0.93 (+0.1)	0.92 (+0.12)

## Data Availability

This study did not report any open-source data.

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
