# Peer review of "Enhancing Real-World Fall Detection Using Commodity Devices: A Systematic Study"

_sensors, 2025, doi:10.3390/s25175249_

Round 1

Reviewer 1 Report

Comments and Suggestions for Authors The research focuses on the integration of multi-sensor data, specifically accelerometer and gyroscope data from the wrist and hip, to improve fall detection performance on commodity devices, particularly in real-world environments. The study fills a critical gap between high-performing offline models and their often-poor real-world deployment performance. The authors emphasize the importance of deployability on low-cost devices and real-time feedback-based adaptation, making this work highly relevant for practical geriatric care and ubiquitous health monitoring.

Compared to prior work, this study evaluates both offline and real-world performance, using only commodity devices and integrating user feedback (TP/FP data) to retrain models (MLMS-II), showing improved generalizability and performance, especially with older adults. The paper explicitly evaluates sensor placement combinations, identifying that dual accelerometer data (wrist + hip) outperforms more complex sensor combinations.

Suggestions for methodology improvement include data imbalance mitigation, sample size diversity, model selection justification, and sensor placement optimization. The Transformer model performed well, but the paper could benefit from a clearer justification for not using lightweight architectures better suited for resource-constrained devices. Additionally, inference latency and resource consumption metrics could be reported. Sensor placement optimization could involve further quantifying ergonomic/usability aspects or investigating new sensor placements.

The conclusions are supported by quantitative metrics (F1-scores, precision/recall) and user feedback from both young and older adult participants. The staged evaluation provides strong triangulation, and the retrained model (MLMS-II) clearly demonstrates reduced false positives without a loss in recall, validating the main hypothesis.

The paper provides a thorough literature review, including both recent and foundational work, and effectively positions the contribution within this context. Some references are fairly recent (2023-2025), ensuring relevance. However, a few important works on lightweight edge inference (e.g., TinyML for HAR or federated learning in health monitoring) might be considered for broader context.

Potential future work includes personalized model tuning, federated learning for privacy-preserving personalization, and more extensive older adult studies.

Author Response

Comments 1: The Transformer model performed well, but the paper could benefit from a clearer justification for not using lightweight architectures better suited for resource-constrained devices. Additionally, inference latency and resource consumption metrics could be reported.

Response 1: Thank you for pointing this out. We agree with your comment. In our previous study, we conducted a comparative analysis of different architectures, including LSTM, CNN-LSTM, and Transformer. Among these, the Transformer model achieved the best performance. This comparative study has already been described in the manuscript (Section 4.3), and the study can be accessed at the following link:

Haque, S.T.; Debnath, M.; Yasmin, A.; Mahmud, T.; Ngu, A.H.H. Experimental Study of Long Short-Term Memory and Transformer Models for Fall Detection on Smartwatches. Sensors 2024, 24, 6235. https://doi.org/10.3390/s24196235

Comments 2: Sensor placement optimization could involve further quantifying ergonomic/usability aspects or investigating new sensor placements.

Response 2: Thank you for this valuable suggestion. While our current work focuses on commodity-based devices and only considers devices that the user is already using, quantifying ergonomic and usability aspects or investigating new sensor placements could provide additional insights. We will consider incorporating this direction in our future work.

Comments 3: The paper provides a thorough literature review, including both recent and foundational work, and effectively positions the contribution within this context. Some references are fairly recent (2023-2025), ensuring relevance. However, a few important works on lightweight edge inference (e.g., TinyML for HAR or federated learning in health monitoring) might be considered for broader context.

Response 3: We thank the reviewer for this thoughtful suggestion. We agree that TinyML and federated learning represent promising directions in the broader area of on-device and privacy-preserving human activity recognition. However, our study specifically targets commodity smartwatches and smartphones as deployment platforms. These devices already support TensorFlow Lite (TFLite) inference with significantly greater computational resources than the microcontroller-class hardware typically considered in TinyML research. As such, we did not position TinyML as directly relevant to our scope, since our focus is not on ultra-constrained microcontroller (MCU) devices but rather on validating multi-sensor fusion and real-world performance of fall detection models on WearOS- and Android-based devices.

Similarly, federated learning primarily addresses distributed and privacy-preserving model training across multiple users. While this is indeed complementary to fall detection research, our work centers on sensor modality fusion, real-time deployment, and model refinement through feedback. Incorporating federated learning considerations would require a separate training infrastructure and data distribution framework, which falls outside the objectives of this paper.

Comments 4: Potential future work includes personalized model tuning, federated learning for privacy-preserving personalization, and more extensive older adult studies.

Response 4: Thank you for your insightful comment. We agree that personalized model tuning, federated learning for privacy-preserving personalization, and more extensive studies involving older adults are promising directions. While these aspects are beyond the scope of our current work, we recognize their potential to further enhance the applicability and robustness of our approach. We have included them in the Future Work section of the Conclusion.

Reviewer 2 Report

Comments and Suggestions for Authors

The manuscript incorporates additional sensor modalities based on accelerometer data from the hip location and re-trains the model using feedback data provided by 10 participants during real-time testing, resulting in a significant performance improvement. Overall, the work is relatively comprehensive. I recommend further refinement in the following aspects:

(1) For tables, please strictly follow the three-line table format. The font alignment within tables should be consistent. For example, in Table 1, some table contents are top-aligned, some are center-aligned, and others are bottom-aligned.

(2) All tables and figures should have clear lines, and the axes should be properly labeled. For example, in Figure 3, the horizontal and vertical axes have no titles, and the axis values are difficult to read.

(3) In Equations (1), (2), and (3), the key letters should be clearly explained to indicate their specific meaning.

Author Response

Comments 1: For tables, please strictly follow the three-line table format. The font alignment within tables should be consistent. For example, in Table 1, some table contents are top-aligned, some are center-aligned, and others are bottom-aligned.

Response 1: We thank the reviewer for pointing this out. In the revised manuscript, we have reformatted all tables to strictly follow the three-line table format. We have also ensured that the font style and alignment are consistent across all table entries, including Table 1.

Comments 2: All tables and figures should have clear lines, and the axes should be properly labeled. For example, in Figure 3, the horizontal and vertical axes have no titles, and the axis values are difficult to read.

Response 2: We thank the reviewer for this observation. All figures and tables have been revised to ensure clear and consistent formatting. Specifically, axis titles and units have been added, tick labels have been adjusted for readability, and table/figure lines have been standardized for clarity. Figure 3 and other figures now clearly display axis labels and values.

Comments 3: In Equations (1), (2), and (3), the key letters should be clearly explained to indicate their specific meaning

Response 3: Thank you for your helpful comment. We agree that the notation in Equations (1), (2), and (3) should be clearly explained. In the revised manuscript, we have added detailed explanations of the key letters immediately following the equations to clarify their specific meaning.

Reviewer 3 Report

Comments and Suggestions for Authors

This paper presents a systematic study on enhancing real-world fall detection using commodity devices, such as smartphones and smartwatches. The paper addresses a practical problem, and its main strengths include real-world evaluation, comparison of sensor configurations and locations, and practical implementation on low-cost hardware. The authors investigate the impact of combining accelerometer and gyroscope data from different sensor locations and using user feedback to improve model performance. A Transformer-based neural network is trained offline, deployed as a TensorFlow Lite model, and further refined using user feedback data to reduce false positives.

However, there are some shortcomings in the paper that should be addressed.

The paper does not present a new method but only provides a systematic comparison of existing methods. In addition, there is no comparison with existing state-of-the-art systems for fall detection.

It is not clear why transformers are used as an architecture and no comparison is made with simpler models more suitable for real-time operation on resource-constrained devices.

There is not enough data on real experiments and different scenarios in which falls were tested. The generalizability of the results is limited because the number of participants in the experiments is relatively small.

A clear discussion of what influenced the number of false positives is lacking.

For practical application in the elderly, the number of false alarms and missed falls is equally important and this should be presented more clearly.

Results for F1-score, precision, and recall are presented, but these metrics are not clearly linked to practical implications. For example, how a given precision or recall value translates to real-world usability.

The stability of the model over time, i.e. the adaptation of the model to changes in user behavior and habits, is not considered.

The equations in Section 4.3 are standard, but not all variables and symbols are defined.

Correct the sentence in lines 112-115.

Author Response

Comments 1: The paper does not present a new method but only provides a systematic comparison of existing methods. In addition, there is no comparison with existing state-of-the-art systems for fall detection.

It is not clear why transformers are used as an architecture and no comparison is made with simpler models more suitable for real-time operation on resource-constrained devices.

Response 1: Thank you for raising this important point. We agree that it is essential to justify the choice of model architecture and provide comparisons with alternative approaches. In our prior work, we performed a detailed comparative analysis of multiple deep learning architectures, including LSTM, CNN-LSTM, and Transformer models, specifically in the context of fall detection. The results showed that the Transformer consistently outperformed the other models in terms of accuracy and robustness, which is why we adopted it for the present study. This comparative evaluation is summarized in Section 4.3 of the manuscript, and the full details can be found in our earlier publication:

Haque, S.T.; Debnath, M.; Yasmin, A.; Mahmud, T.; Ngu, A.H.H. Experimental Study of Long Short-Term Memory and Transformer Models for Fall Detection on Smartwatches. Sensors, 2024, 24, 6235. https://doi.org/10.3390/s24196235

In the current work, our focus is on providing a systematic evaluation of real-time fall detection performance under real-world conditions, using the Transformer as the most effective model identified in our previous study.

Comments 2: A clear discussion of what influenced the number of false positives is lacking.

Response 2: Thank you for this comment. We agree that clarifying the factors contributing to false positives is important. In our study, we observed that false positives were primarily influenced by individual differences in gait and motion signatures. Everyday activities such as abrupt arm movements, quick changes in posture, or vigorous ADLs (e.g., exercising or rapid sitting/standing) sometimes mimic the signal characteristics of falls, leading to misclassifications. Since each user has unique movement dynamics, the same activity may appear benign for one participant but trigger a false alarm for another. To address this, we incorporated user feedback data, allowing the system to learn from these misclassifications and gradually adapt to the individual’s motion patterns. This explanation has been added to the Discussion (Section 7) to make the practical implications clearer.

Comments 3: For practical application in the elderly, the number of false alarms and missed falls is equally important and this should be presented more clearly.

Response 3: Thank you for your valuable comment. We fully agree that both false alarms and missed falls are critical factors for practical deployment in elderly populations. To address this, we have clarified our treatment of false negatives (FNs) in Section 6.1. Specifically, if the model failed to recognize a fall, users were able to manually report the event using the “FELL” button on the app’s home screen, ensuring that such incidents were not ignored but properly logged as FNs. In our real-world evaluation, these user-reported FNs were treated with the same importance as true positives (TPs). Both FN and TP samples were merged and used in retraining, which allowed the model to continuously learn from its mistakes and improve over time. This feedback loop directly addresses the concern of missed falls while also contributing to reducing false alarms, thereby strengthening the system’s reliability for real-world use.

Comments 4: Results for F1-score, precision, and recall are presented, but these metrics are not clearly linked to practical implications. For example, how a given precision or recall value translates to real-world usability.

Response 4: Thank you for raising this important point. We agree that linking performance metrics to real-world usability is crucial. To address this, we have expanded the discussion in Section 5.1, where we explain how the reported results translate into practical implications. Specifically, recall directly relates to the system’s ability to detect true falls, which is critical for ensuring user safety, while precision indicates how often the detected falls are indeed real falls, which is important for minimizing false alarms and preventing alarm fatigue. The F1-score provides a balanced perspective of both aspects. By adding this explanation, we ensure that the reported metrics are contextualized for real-world deployment and usability.

Comments 5: The stability of the model over time, i.e. the adaptation of the model to changes in user behavior and habits, is not considered.

Response 5: Thank you for your valuable comment. You are correct that long-term stability and adaptation of the model to evolving user behavior an important aspects of real-world deployment. In our study, we conducted only a single round of personalization, where user feedback data was incorporated to improve the model’s performance on individual activity patterns. While further continuous adaptation and stability analysis over time are indeed possible, they are beyond the scope of this work. The primary objectives here were (i) to investigate whether using multiple sensor locations (wrist and hip) improves the model’s ability to distinguish falls from ADLs, and (ii) to demonstrate that incorporating user feedback data can make the model more precise. Future work will focus on more extensive longitudinal personalization and continuous adaptation to user behavior.

Comments 6: The equations in Section 4.3 are standard, but not all variables and symbols are defined.

Response 6: Thank you for your helpful comment. We agree that the notation in Equations (1), (2), and (3) in Section 4.3 should be clearly explained. In the revised manuscript, we have added detailed explanations of the key letters immediately following the equations to clarify their specific meaning.

Comments 7: Correct the sentence in lines 112-115.

Response 7: Thank you for pointing this out. We have corrected the sentence in lines 112–115 for clarity and readability.

Reviewer 4 Report

Comments and Suggestions for Authors

The manuscript presents research aimed at improving real-world fall detection performance on low-cost commodity devices by combining accelerometer and gyroscope data from wrist and hip sensors, with a focus on older adults. The authors conduct systematic experiments using a Transformer-based model, comparing single-sensor setups with multi-sensor configurations. They evaluate the best offline model in both controlled and uncontrolled environments. The study demonstrates that combining accelerometer data from two locations improves performance, and that feedback-based retraining further boosts precision without sacrificing recall. In addition, a small-scale evaluation with older adults shows promising generalizability.

The manuscript is generally clear and readable, with well-presented tables and methodology. However, I have minor comments, which are listed below.

1. Abstract

If I’m not mistaken, the wrist location is not mentioned in Line 7, where only the hip location appears. This could confuse the reader, since both locations are used in the study.

2. Related work

The problem of the limited amount of available fall data (Lines 78-79) feels somewhat overstated. While falls are indeed rare events, there are existing datasets and publications that have not been mentioned in the manuscript. A broader review of the literature and a comparison with state-of-the-art research would strengthen the contribution. Below are several references the authors may wish to consider including:

  • The FARSEEING real-world fall repository (https://farseeingresearch.eu/) and the corresponding paper: DOI: 10.1186/s11556-016-0168-9;
  • Aziz, O., Musngi, M., Park, et al. (2016) A comparison of accuracy of fall detection algorithms (threshold-based vs. machine learning) using waist-mounted tri-axial accelerometer signals from a comprehensive set of falls and non-fall trials. Medical & Biological Engineering & Computing, 55(1), 45–55. DOI: 10.1007/s11517-016-1504-y;
  • Bagalà, F., Becker, C., Cappello, et al. (2012) Evaluation of Accelerometer-Based Fall Detection Algorithms on Real-World Falls. PLoS ONE 7(5): e37062. DOI:10.1371/journal.pone.0037062;
  • Broadley, R.W.; Klenk, J.; Thies, S.B.; Kenney, L.P.J.; Granat, M.H. Methods for the Real-World Evaluation of Fall Detection Technology: A Scoping Review. Sensors 2018, 18, 2060. DOI: 10.3390/s18072060;
  • Harari, Y., Shawen, N., Mummidisetty, C.K. et al. A smartphone-based online system for fall detection with alert notifications and contextual information of real-life falls. NeuroEngineering and Rehabilitation 18, 124 (2021). DOI: 10.1186/s12984-021-00918-z;
  • Ruiz-Garcia, J. C., et al. CareFall: Automatic Fall Detection through Wearable Devices and AI Methods, arXiv:2307.05275. DOI: 10.48550/arXiv.2307.05275.

3. User feedback integration

A brief explanation would be helpful as to why the authors did not take false negatives into account when retraining the model (Lines 432-433).

4. Discussion section

The Discussion section is missing, which I consider to be a drawback. This is where the reader would normally expect to find a detailed explanation of the pros and cons of the proposed approaches, as well as a numerical comparison of the results with those obtained by other teams.

In conclusion, this work is highly relevant to researchers and practitioners interested in real-world wearable-based fall detection, and I recommend it for publication after minor revisions.

Author Response

Comments 1: If I’m not mistaken, the wrist location is not mentioned in Line 7, where only the hip location appears. This could confuse the reader, since both locations are used in the study.

Response 1: Thank you for bringing this to our attention. We agree that the omission could confuse. We have revised the sentence in Line 7 to explicitly mention both the wrist and hip locations to ensure clarity.

Comments 2: The problem of the limited amount of available fall data (Lines 78-79) feels somewhat overstated. While falls are indeed rare events, there are existing datasets and publications that have not been mentioned in the manuscript. A broader review of the literature and a comparison with state-of-the-art research would strengthen the contribution. Below are several references the authors may wish to consider including:

The FARSEEING real-world fall repository (https://farseeingresearch.eu/) and the corresponding paper: DOI: 10.1186/s11556-016-0168-9;

Aziz, O., Musngi, M., Park, et al. (2016) A comparison of accuracy of fall detection algorithms (threshold-based vs. machine learning) using waist-mounted tri-axial accelerometer signals from a comprehensive set of falls and non-fall trials. Medical & Biological Engineering & Computing, 55(1), 45–55. DOI: 10.1007/s11517-016-1504-y;

Bagalà, F., Becker, C., Cappello, et al. (2012) Evaluation of Accelerometer-Based Fall Detection Algorithms on Real-World Falls. PLoS ONE 7(5): e37062. DOI:10.1371/journal.pone.0037062;

Broadley, R.W.; Klenk, J.; Thies, S.B.; Kenney, L.P.J.; Granat, M.H. Methods for the Real-World Evaluation of Fall Detection Technology: A Scoping Review. Sensors 2018, 18, 2060. DOI: 10.3390/s18072060;

Harari, Y., Shawen, N., Mummidisetty, C.K. et al. A smartphone-based online system for fall detection with alert notifications and contextual information of real-life falls. NeuroEngineering and Rehabilitation 18, 124 (2021). DOI: 10.1186/s12984-021-00918-z;

Ruiz-Garcia, J. C., et al. CareFall: Automatic Fall Detection through Wearable Devices and AI Methods, arXiv:2307.05275. DOI: 10.48550/arXiv.2307.05275.

Response 2: We thank the reviewer for pointing out these important references. We are aware of these datasets and publications, and agree that they represent valuable contributions to the field. However, these datasets were not suitable for the objectives of our work, which specifically focuses on multimodal (accelerometer + gyroscope) signals collected simultaneously from both the wrist and the hip. The reasons are as follows:

FARSEEING repository (Klenk et al., 2016; Bagalà et al., 2012): While it provides invaluable real-world fall data, the available recordings are highly heterogeneous (different devices, placements, and sampling frequencies). Crucially, most records consist of waist-only accelerometer data, with limited availability of gyroscope information and no simultaneous wrist–hip recordings.

Aziz et al., 2017: This study employed a waist-mounted accelerometer only. No gyroscope data and no wrist signals were available, making it unsuitable for our dual-location, multimodal framework.

Broadley et al., 2018 (scoping review): This review summarizes evaluation methods but does not provide new datasets. The majority of the studies it surveys rely on single-location sensors (typically waist) or ambient/vision-based systems, again not aligned with our sensor configuration.

Harari et al., 2021: Their prospective real-world work is based on a commodity smartphone accelerometer/gyroscope, typically carried at the waist or in a pocket. It does not include simultaneous wrist data, and sensor location varied considerably among participants.

Ruiz-García et al., 2023 (CareFall): This recent work targets wrist-worn smartwatches and evaluates wrist-only data from public datasets. While valuable, it lacks complementary hip sensor recordings, which are central to our investigation of dual-location fusion.

In summary, while these datasets and studies significantly advance fall detection research, none of them provide the specific combination of wrist and hip accelerometer + gyroscope data required to evaluate our proposed method. We have added clarifications in the Discussion section of the manuscript to make this point more explicit and to acknowledge these works appropriately.

Comments 3: A brief explanation would be helpful as to why the authors did not take false negatives into account when retraining the model (Lines 432-433).

Response 3: Thank you for your comment. If the model fails to detect a fall, the user can manually press the “FELL” button on the application’s home screen, ensuring that the event is recorded as a false negative (FN). We have added this clarification in the revised manuscript in section 6.1.

Comments 4: The Discussion section is missing, which I consider to be a drawback. This is where the reader would normally expect to find a detailed explanation of the pros and cons of the proposed approaches, as well as a numerical comparison of the results with those obtained by other teams.

Response 4: We thank the reviewer for this helpful suggestion. In the revised manuscript, we have now included a dedicated Discussion section. This section elaborates on the advantages and limitations of our proposed approach, provides a critical reflection on the results, and offers a comparative perspective with related state-of-the-art methods. We believe this addition substantially strengthens the manuscript and addresses the reviewer’s concern.